# Osteomyelitis Possibly Caused by Exfoliation of Primary Teeth in a Patient with Osteopetrosis

**DOI:** 10.3390/children9121894

**Published:** 2022-12-02

**Authors:** Rena Okawa, Yusuke Yokota, Yoshihiro Morita, Yusuke Mikasa, Kazuhiko Nakano

**Affiliations:** 1Department of Pediatric Dentistry, Osaka University Graduate School of Dentistry, Osaka 565-0871, Japan; 2First Department of Oral and Maxillofacial Surgery, Osaka University Graduate School of Dentistry, Osaka 565-0871, Japan; 3Second Department of Oral and Maxillofacial Surgery, Osaka University Graduate School of Dentistry, Osaka 565-0871, Japan

**Keywords:** osteomyelitis, osteopetrosis, primary teeth

## Abstract

Osteopetrosis is a rare inherited skeletal disease characterized by increased density in the bones and can be detected by radiographs. Sclerosis makes the jaw bones extremely susceptible to infections, osteomyelitis, and fractures. Here, we report a case of osteomyelitis possibly caused by spontaneous exfoliation of primary teeth in a patient with osteopetrosis. A 2 years and 9 months (2Y9M)-old boy with osteopetrosis was referred to our clinic for oral management. Only four primary central incisors had erupted, and they all exhibited hypoplasia. The mandibular right central primary incisor suddenly became exfoliated at 4 years and 1 month. The mandibular right lateral primary incisor also became exfoliated at 4 years and 3 months, soon after eruption, and the mandibular left central primary incisor became exfoliated at 4 years and 5 months. Subsequently, we confirmed the eruption of calcified tissue at 4 years and 9 months in the location where the mandibular right lateral primary incisor had become exfoliated. The patient was admitted to the pediatrics clinic for mandibular cellulitis at 5 years and 2 months, then referred to our clinic for the management of osteomyelitis. The patient’s acute inflammation was reduced by repeated irrigation and the administration of antibiotics; the inflammation gradually became chronic. When treating patients with osteopetrosis, dentists and oral surgeons should prioritize infection control in the jaw, periodic assessment of dental eruption, and the maintenance of oral hygiene.

## 1. Introduction

Osteopetrosis (marble bone disease) is a rare inherited skeletal disease that was first reported by Albers-Schönberg in 1904 [1]. It is characterized by increased density in the bones, and it can be detected by radiographs [1,2]. Osteoblast dysfunction causes an inability to reabsorb and remodel bone [1,2]. The estimated incidence of osteopetrosis is 0.6 per 100,000 live births in Japan, which is similar to the rates in other countries [3]. Osteoporosis is classified into three clinical types based on severity and onset: early-onset and severe neonatal/infant, moderate-intermediate, and late-onset (adult) milder [1,2,3]. The early-onset and severe neonatal/infant type and the moderate-intermediate type are inherited in an autosomal recessive manner, whereas the late-onset milder type is inherited in an autosomal dominant manner [1,2,3]. In the early-onset and severe neonatal/infant type, very young patients exhibit symptoms of bone marrow dysfunction (e.g., anemia, bleeding tendency, and susceptibility to infection), as well as clinically significant delayed growth, cranial nerve damage, hydrocephalus, and hypocalcemia; long-term survival may be impossible [1,2,3]. In the moderate-intermediate type, patients exhibit fractures, osteomyelitis, and dental abnormalities during childhood [1,2,3]. In the milder late-onset type, patients exhibit symptoms such as bone fractures, osteomyelitis, and facial paralysis in adulthood [1,2,3].

Dental manifestations of osteopetrosis include defective resorption-related delayed tooth eruption, tooth absence, unerupted and malformed teeth, enamel and dentinal hypoplasia, abnormal pulp chambers, a tendency toward early decay, periodontal membrane defects, a thickened lamina dura, and dental caries-induced early tooth loss [4,5,6,7,8,9,10]. Sclerosis makes the jaw bones extremely susceptible to infections, osteomyelitis, and fractures [2,4,5,6,7,8,9,10]. Extractions are usually difficult, and bony socket healing is poor, leading to irregular alveolar ridges [2,4,5,6,7,8,9].

Here, we report a case of osteomyelitis caused by spontaneous exfoliation of primary teeth in a patient with osteopetrosis. Informed consent was obtained from the patient’s parents for the publication of this case report and the accompanying images.

## 2. Case Presentation

A 2 years and 9 months (2Y9M)-old Japanese boy was referred to the Pediatric Dentistry Clinic of Osaka University Dental Hospital from the pediatric dental clinic of a children’s hospital for oral management. The patient had previously been diagnosed with osteopetrosis. He had undergone hydrocephalus surgery at the ages of 1 and 2 years, and he had developed optic nerve atrophy and hearing impairment; however, he had no history of fractures. Two mandibular primary incisors had erupted at 4 months after birth. Four mandibular and maxillary primary central incisors were erupted at the time of the first examination. He exhibited mental retardation, which hindered the dental examination. Enamel transparency was poor, and enamel hypomineralization was evident (Figure 1a). Periapical radiographic examination revealed multiple impacted teeth (Figure 1b). We decided to observe his tooth eruption while evaluating his oral hygiene. We also referred the patient to the Department of Oral and Maxillofacial Surgery, where the clinicians planned to wait for growth prior to performing further examinations (e.g., computed tomography [CT]) under sedation.

The mandibular right central primary incisor suddenly became exfoliated at 4 years and 1 month. According to the patient’s guardians, there were no external factors, such as dental trauma. The root of the exfoliated tooth was not absorbed or broken (Figure 2a). The exfoliated tooth was analyzed using a micro-CT device (R_mCT2; Rigaku^®^, Tokyo, Japan) to assess its morphology (Figure 2b). The results revealed poorly marginated enamel and dentin, along with perforation of the pulp chamber.

Periapical radiographic examination did not reveal a residual root of the mandibular right central primary incisor, although the germ of the permanent successor was evident (Figure 3a). The socket was covered with gingiva (Figure 3a). The mandibular right lateral primary incisor also became exfoliated at 4 years and 3 months, soon after eruption. According to the patient’s guardians, the tooth had been loose since it erupted 1 month prior (Figure 3b). The mandibular left central primary incisor also became exfoliated at 4 years and 5 months. Subsequently, we confirmed the eruption of calcified tissue at 4 years and 9 months in the location where the mandibular right lateral primary incisor had become exfoliated at 4 years and 3 months (Figure 3c,d).

The patient was admitted to the Department of Pediatrics for treatment of mandibular cellulitis at 5 years and 2 months (WBC: 13,500/μL and CRP: 3.78 mg/dL). He exhibited repeated fevers for 1 month, along with other clinical findings around the mandible (e.g., redness and swelling). CT images of the submental area revealed an obvious abscess (Figure 4).

Ceftriaxone was administered, and incision and drainage were performed. Viridans streptcocci and *Tannerella forsythia* were detected in the affected region. After resolution of the inflammation, the patient was referred to our clinic for the management of osteomyelitis. Alveolar bone around the mandibular right central incisor was exposed, and calcified tissue around the mandibular right lateral incisor exhibited severe mobility. With the patient under local anesthesia, an oral surgeon performed extensive alveolar bone removal and curettage of infected granulation tissue in the mandibular right central incisor region, extraction of calcified tissue in the mandibular right lateral incisor region, and gingival suturing. Amoxicillin was also administered. Histopathological examination revealed that the calcified tissue comprised a tooth-like structure (Figure 5a,b). The tooth-like tissue was undergoing root formation, and no inflammation was observed in the dental papilla. However, inflammatory findings were evident in the coronal pulp region that was exposed to the oral cavity (Figure 5c).

We presumed that the remaining dental papilla cells in the exfoliated mandibular right lateral incisor formed a vital calcified tissue with a root-like appearance. We determined that the patient’s osteomyelitis had been caused by an infection in the alveolar bone socket of the exfoliated mandibular right central incisor. We decided to repeat the irrigation treatment. The patient’s inflammation was reduced and gradually became chronic (Figure 6).

Unfortunately, mandibular swelling recurred after a few months. The patient’s pediatrician administered meropenem when acute inflammation became evident. Aggressive sequestrotomy under general anesthesia was difficult because of the risk of mandibular fracture. We decided to perform more frequent irrigation and oral hygiene control, and we referred the patient to the Department of Oral Surgery at a general hospital near his home. At 6 years and 5 months, the number of fistulae had increased, but there were no signs of acute inflammation (Figure 7).

## 3. Discussion

Osteopetrosis is accompanied by many oral problems: delayed or failed tooth eruption, the absence of some teeth, tooth malformation, enamel and dentinal hypomineralization, periodontal membrane defects, and thickened lamina dura [4,5,6,7,8,9,10]. Delayed or failed tooth eruption is caused by an increasingly insufficient supply of nutrients to the developing tooth germ, which results from increased bone density [9,10,11,12]. Our patient exhibited this finding: only four primary central incisors erupted within 1 year after birth. A tooth eruption might have been possible before the progression of osteosclerosis. However, the fenestration and protrusion of impacted teeth are difficult for patients with osteopetrosis, in whom eruption cannot be expected because of ankylosis in the cementum interface along with the lack of an alveolar ligament [12]. Surgical management is accompanied by high risks of osteomyelitis and bone fractures [2,6,7,8,9,10,11]. Dentures are fabricated for patients with osteopetrosis who exhibit multiple impacted teeth [7,13,14]. Such fabrication could not be performed for our patient because of his severe mental retardation. Frequent denture adjustment is recommended to prevent the development of ulcers under dentures [7]. Osteomyelitis in patients with osteopetrosis can be caused by infections that occur after tooth extraction [12,15,16,17]. Osteomyelitis can also be caused by impaired alveolar socket repair after delayed wound healing because of reduced blood supply to the jaw [2,4,5,6,7,8,9,10]. When surgical interventions are performed, antibiotic prophylaxis is recommended to prevent infection and osteomyelitis related to the reduced periosteal blood supply [6,8].

There have been two reports concerning the spontaneous exfoliation of primary teeth in a patient with osteopetrosis [13,18]. In one patient, this phenomenon was caused by poor fibrous attachment [13]. After the exfoliation of a primary incisor, a tooth-like structure emerged into the oral cavity in our patient. The underdeveloped dental papilla cells and underdeveloped Hertwig’s root sheath might easily become detached from the calcified portion of a tooth and remain in the alveolus, subsequently developing into a tooth-like structure that erupts months or years later [19]. Histopathological analysis was performed to investigate whether the infection was derived from this tooth-like structure or from the exfoliation socket of the primary central incisor. Although the pulp of the tooth-like structure exposed to the oral cavity exhibited bacterial infection, the root pulp appeared vital. We determined that the infection had originated in the exfoliation socket of the primary central incisor. The primary central incisor was decayed or ablated because of enamel hypoplasia, and the pulp was exposed to the oral cavity. Infection of the pulp with oral bacteria resulted in periodontitis. Then, the tooth became exfoliated because of alveolar bone resorption. In patients with tooth exfoliation, gingival healing should be assessed frequently, antibiotics should be administered as necessary, and the infected tissues should be curetted.

In our patient, acute inflammation was reduced once by repeated irrigation and the administration of antibiotics. However, osteomyelitis subsequently appeared in other sites in a sequential manner. There are few published reports concerning the successful treatment of osteomyelitis in children [20,21,22,23,24]. Surgical intervention with drug therapy (i.e., systemic administration of antibacterial agents) is often used to manage osteomyelitis [6,22,23,24]. Local sequestrectomy of the affected region is sometimes performed [12,16,20]. However, osteomyelitis recurs because poor bone blood flow hinders antibiotic penetration of affected tissue, and it is difficult to distinguish sclerotic from necrotic bone during sequestrum removal [22]. In patients with multiple instances of osteomyelitis, corticotomy is sometimes performed [17,22]. Generally, radical mandibular resections (e.g., resections that require reinforcement with titanium plates) are not preferred because they carry a risk of mandibular growth inhibition [25,26]. We considered aggressive surgery a final treatment method due to the age of the patient, general condition, and mental retardation.

## 4. Conclusions

In patients with osteopetrosis, osteomyelitis remains unresolved indefinitely [21]. Even tooth eruption can result in serious infection [16,27]. Infection control in the jaw should be a priority for dentists and oral surgeons [5,16]. For pediatric dentists, the fundamental treatment of osteopetrosis involves frequent assessment of dental eruption and the maintenance of oral hygiene [8,23,24,27].

## Figures and Tables

**Figure 1 children-09-01894-f001:**
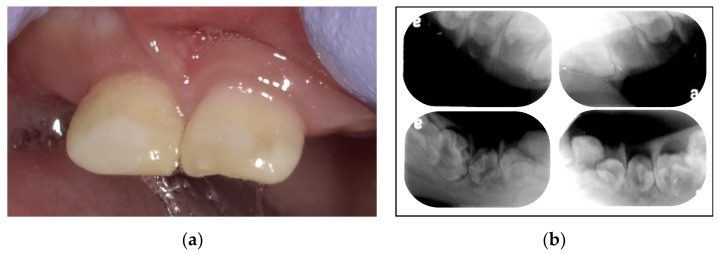
(**a**) Intraoral photograph acquired at the age of 3 years and 2 months. (**b**) Periapical radiograph images acquired at the age of 3 years and 2 months.

**Figure 2 children-09-01894-f002:**
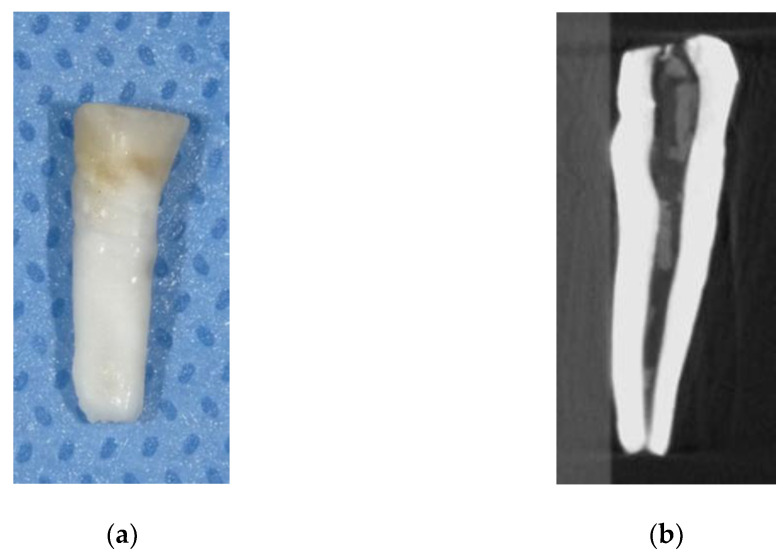
(**a**) Mandibular right central primary incisor, exfoliated at the age of 4 years and 1 month. (**b**) micro-CT analysis of the exfoliated tooth.

**Figure 3 children-09-01894-f003:**
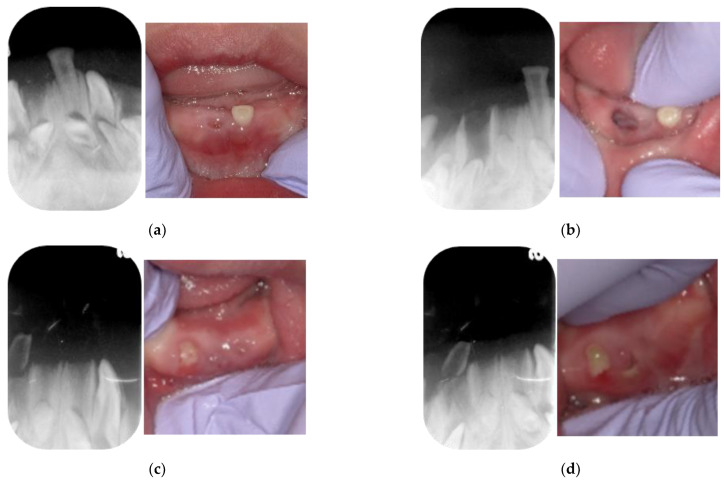
Periapical radiographs and intraoral photographs acquired at the ages of (**a**) 4 years and 1 month, (**b**) 4 years and 2 months, (**c**) 4 years and 8 months, and (**d**) 4 years and 11 months.

**Figure 4 children-09-01894-f004:**
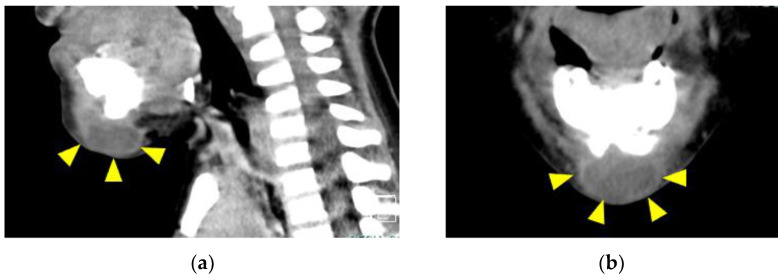
CT images: (**a**) sagittal section and (**b**) coronal section. An abscess cavity was evident in the submental area (yellow arrowheads).

**Figure 5 children-09-01894-f005:**
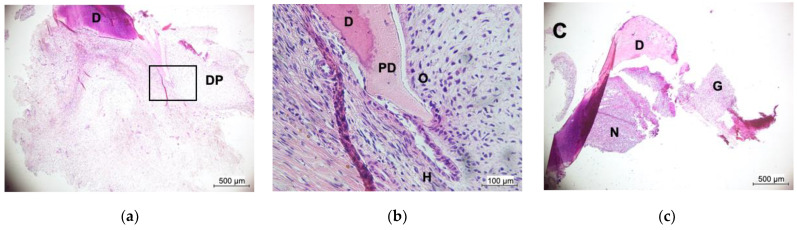
Histopathological examination of the mandibular right primary lateral incisor (hematoxylin-eosin staining). (**a**) Lower magnification of the root apex region. (**b**) A magnified view of the area in (**a**) enclosed by a square. (**c**) Lower magnification of pulp in the crown area. DP: dental papilla, D: dentin, PD: pre-dentin, O: odontoblast, H: Hertwig’s root sheath, N: neutrophil, and G: granulation tissue.

**Figure 6 children-09-01894-f006:**
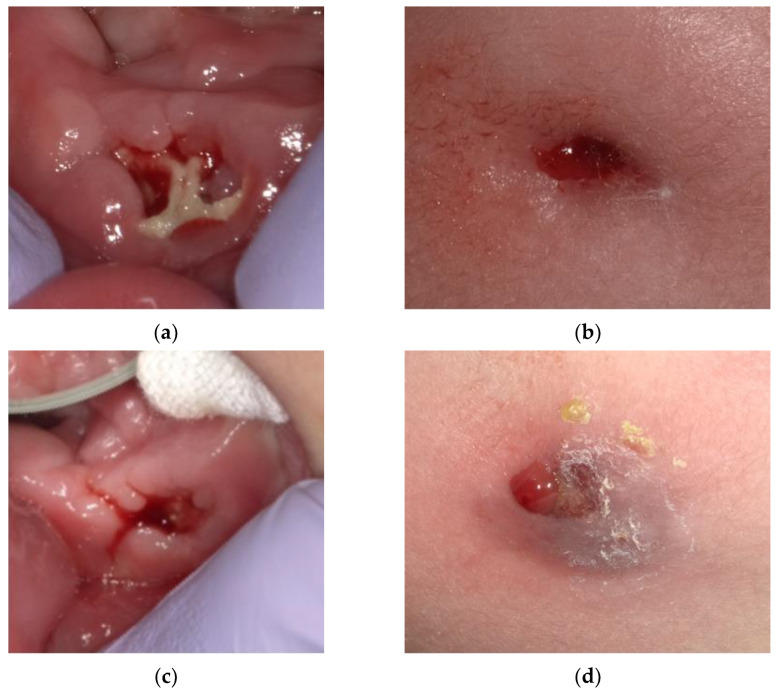
(**a**,**b**) Intra- and extraoral photographs acquired at the age of 5 years and 4 months. (**c**,**d**) Intra- and extraoral photographs acquired at the age of 5 years and 5 months.

**Figure 7 children-09-01894-f007:**
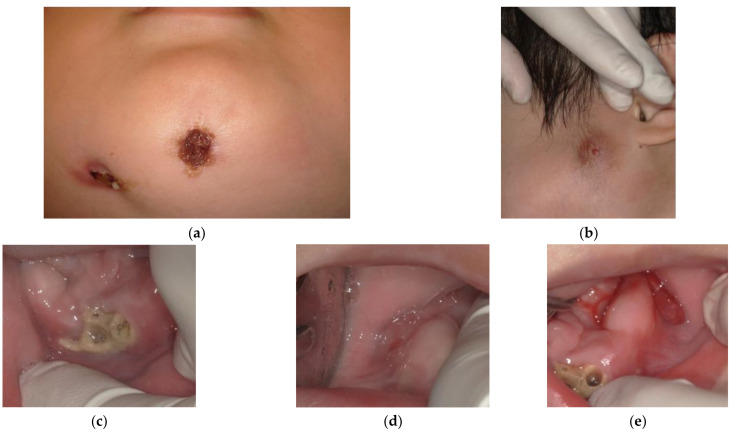
Extra- and intraoral photographs acquired at the age of 6 years and 5 months: (**a**) mandibular, (**b**) left side of face, (**c**) mandibular, (**d**) mandibular right region, and (**e**) mandibular left region.

## Data Availability

Not applicable.

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
