# Peer review of "Osteomyelitis Possibly Caused by Exfoliation of Primary Teeth in a Patient with Osteopetrosis"

_children, 2022, doi:10.3390/children9121894_

Round 1

Reviewer 1 Report

  1. There is importance and soundness of the presentation of this paper and is very useful for clinicians to learn how to recognise and treat this symptoms.
  2. Also figures and tables are very well presented.

Author Response

We thank the reviewer for the effort taken to review our paper and for these supportive comments.

Reviewer 2 Report

Dear authors, well done on the good work, research and case presentation.

Author Response

(The authors gave the same response as above.)

Reviewer 3 Report

Congratulations on the authors for a good case report, I would like to highligth some point in order to helkp you with the study:

In the introduction I feel there is few references about the mechanism of the bone sclerosis and the enamel and dentin disorders secondary to the osteopetrosis.

[Sekerci AE, Sisman Y, Ertas ET, Sahman H, Aydinbelge M. Infantile malignant osteopetrosis: report of 2 cases with osteomyelitis of the jaws. J Dent Child (Chic). 2012 May-Aug;79(2):93-9. PMID: 22828766.]

I have a question, you say that you determine taht the osteomyelitys was becuase of the infection of the alvolar socket os the deciduos teeth, apart from irrigation it has been demonstrated by the articles taht you mention in your discussion taht there are other treatments with more success, why didn´t you use that? 

Author Response

(Response) We thank the reviewer for the effort taken to review our paper and for these supportive comments. We have added your suggested reference. We have also described in the last of discussion section why other treatments did not apply in this case.